# Kullback–Leibler Divergence of Sleep-Wake Patterns Related with Depressive Severity in Patients with Epilepsy

**DOI:** 10.3390/brainsci13050823

**Published:** 2023-05-19

**Authors:** Mingsu Liu, Jian Jiang, Yu Feng, Yang Cai, Jing Ding, Xin Wang

**Affiliations:** 1Department of Neurology, Zhongshan Hospital, Fudan University, Shanghai 200032, China; 20111210063@fudan.edu.cn (M.L.); feng.yu@zs-hospital.sh.cn (Y.F.); cai.yang@zs-hospital.sh.cn (Y.C.); wang.xin@zs-hospital.sh.cn (X.W.); 2Institute of Neuroscience, Key Laboratory of Primate Neurobiology, CAS Center for Excellence in Brain Science and Intelligence Technology, Chinese Academy of Sciences, Shanghai 200050, China; jiangjian@eegion.com; 3CAS Center for Excellence in Brain Science and Intelligence Technology, Shanghai 200031, China; 4Department of the State Key Laboratory of Medical Neurobiology and MOE Frontiers Center for Brain Science, Institutes of Brain Science, Fudan University, Shanghai 200032, China

**Keywords:** epilepsy, depression, EEG, Kullback–Leibler divergence

## Abstract

(1) Objective: Whether abnormal sleep-wake rhythms were associated with depressive symptoms in patents with epilepsy had remained unclear. Our study aimed to establish relative entropy for the assessment of sleep-wake patterns and to explore the relationship between this index and the severity of depressive symptoms in patients with epilepsy. (2) Methods: We recorded long-term scalp electroencephalograms (EEGs) and Hamilton Depression Rating Scale-17 (HAMD-17) questionnaire scores from 64 patients with epilepsy. Patients with HAMD-17 scores of 0–7 points were defined as the non-depressive group, while patients with scores of 8 or higher were defined as the depression group. Sleep stages were firstly classified based on EEG data. We then quantified sleep-wake rhythm variations in brain activity using the Kullback–Leibler divergence (KLD) of daytime wakefulness and nighttime sleep. The KLD at different frequency bands in each brain region was analyzed between the depression and non-depression groups. (3) Results: Of the 64 patients with epilepsy included in our study, 32 had depressive symptoms. It was found that patients with depression had significantly decreased KLD for high-frequency oscillations in most brain areas, especially the frontal lobe. A detailed analysis was conducted in the right frontal region (F4) because of the significant difference in the high-frequency band. We found that the KLDs at the gamma bands were significantly decreased in the depression groups compared to the non-depression group (KLD_D_ = 0.35 ± 0.05, KLD_ND_ = 0.57 ± 0.05, *p* = 0.009). A negative correlation was displayed between the KLD of gamma band oscillation and HAMD-17 score (r = −0.29, *p* = 0.02). (4) Conclusions: Sleep-wake rhythms can be assessed using the KLD index calculated from long-term scalp EEGs. Moreover, the KLD of high-frequency bands had a negative correlation with HAMD-17 scores in patients with epilepsy, which indicates a close relationship between abnormal sleep-wake patterns and depressive symptoms in patients with epilepsy.

## 1. Introduction

Epilepsy, one of the most common neurological disorders affecting more than 65 million people worldwide, is characterized by spontaneous, unpredictable, and recurrent seizures [1]. Epilepsy and circadian rhythms have reciprocal manners in several physiological processes. Circadian rhythms are part of the internal daily rhythms of nearly all biological functions, and they serve as an adaptive response to cyclical environments such as light, food, temperature, and sleep [2]. In turn, disturbance to the circadian rhythm is usually caused by environmental factors or internal factors of the body, which affect the body clock and eventually lead to sleep rhythm abnormality. People with epilepsy have a higher prevalence of sleep disorders, which are usually characterized by sleep fragmentation, reduced sleep duration, variation in sleep structure, and daytime sleepiness [3,4]. However, the evidence in sleep-wake rhythm in epilepsy is still insufficient, which may be related to the lack of relevant measures and indexes. Non-invasive electroencephalograms (EEGs) are safe for extensive applications because they do not require any invasive procedures. Determining whether a novel assessment index for circadian variation can be established based on scalp EEG is one of our focuses. Kullback–Leibler divergence (KLD) is a measure of divergence change between two states [5], thus, we use KLD of awake and sleep stages from EEG recording to quantify the sleep-wake pattern.

In addition to sleep disturbance, depression affects up to 55% of patients with epilepsy, which may be the strongest predictor of life quality and of high risk for suicide in epilepsy [6]. Recently, there has been accumulating evidence revealing that sleep disturbances may exist years before the first onset of a depressive episode. In a study focused on the relationship between sleep disturbance and subsequent depression, sleep disturbance was considered as a significant risk factor for subsequent depression and persisted for at least 30 years [7]. Abnormalities in circadian rhythms are well observed in adults with depressive disorders and have been linked to core clinical features; furthermore, it has been suggested that the disturbance of sleep-wake rhythms is a possible pathophysiological pathway of depression [8,9]. However, evidence on whether abnormal sleep-wake rhythms are associated with depression symptoms in patients with epilepsy remains unavailable. Therefore, we will investigate the relationship between the KLD index and depressive severity by fitting a linear correlation in patients with epilepsy in this study.

## 2. Methods

### 2.1. Study Participants

Consecutive patients with epilepsy admitted to the neurology department of Zhongshan Hospital in Shanghai, China between January 2016 and June 2020 were included in this retrospective study. Patients were diagnosed with epilepsy according to the diagnostic criteria of the International League Against Epilepsy [10]. Epilepsy type mainly includes generalized epilepsy, focal epilepsy, and an unknown category. The diagnosis of generalized epilepsy is made on clinical grounds (e.g., absence, myoclonic, atonic, tonic and tonic-clonic seizures), supported by the finding of typical interictal EEG discharges. A range of seizure types of focal epilepsy can be seen including focal aware seizures, focal impaired awareness seizures, focal motor seizures, focal non-motor seizures, and focal to bilateral tonic-clonic seizures when the interictal EEG typically shows focal epileptiform discharges. Moreover, focal epilepsy can be classified into temporal lobe epilepsy, frontal lobe epilepsy and occipital lobe epilepsy according to the electroclinical manifestations. The term “unknown epilepsy” is used to denote situations where it is understood that the patient has epilepsy but the clinician is unable to determine if the epilepsy type is focal or generalized because there is insufficient information available. For example, the patient may have had several symmetrical tonic-clonic seizures without focal features and normal EEG recordings. Thus, the onset of the seizure is unknown and the person has an unknown epilepsy type. Long-term EEG and psychological assessment data must be available for all enrolled patients. Subjects meeting the criteria were extracted from the clinical database, which was mainly conducted by Liu and Cai. EEG Evaluation is also required, including integrity, monitoring duration, and interference in addition to seizure events. The investigators were not blind to each other. For all participants, comprehensive data, including clinical characteristics such as age, sex, latency, first diagnosis of depression, type of seizure, and drug administration were collected. Patients with epilepsy had to show either no abnormalities or only hippocampal sclerosis on MRI or CT scans. Participants under 14 years of age were excluded, as were those who had current diagnoses of neurological diseases, or severe systematic disease (e.g., kidney failure, heart failure, respiratory failure, or infectious diseases). Patients who were pregnant or breast-feeding, had a medical disorder, substance abuse, or a history of bipolar I or II, or obsessive compulsive disorder were also excluded from the study.

The neuropsychological assessment was performed right after the EEG examination by one trained specialist according to a standardized protocol especially designed for patients with epilepsy. The 17-item Hamilton Depression Rating Scale (HAMD-17) was used to define depressive severity in patients with epilepsy. Based on the HAMD-17 scores, the depression group was defined as patients with scores of 0–7 points and the depression group was defined as patients with scores of greater than 8 points [11] (Figure 1A). In addition, scores for the Mini-Mental State Examination (MMSE), auditory verbal learning test (AVLT), digital span (DS), symbol digit modalities test (SDMT), and verbal fluency test (VFT) were individually used to assess participants’ global cognitive function, specific verbal memory function, attentional function, executive function, and verbal function, respectively [12,13]. Cognitive and behavioral performance identified by neuropsychological assessments contribute to a number of important comprehensive care programs for patients with epilepsy. This study was approved by the ethics committee of the Zhongshan Hospital, Fudan University.

### 2.2. Data Acquisition

One-day EEG data were recorded using the CMS system (Cadwell Spectrum 32 recording instrument) in accordance with the international 10–20 system. Nineteen surface electrodes (F3, F4, F7, F8, P3, P4, T5, T6, C3, C4, T3, T4, O1, O2, Fp1, Fp2, Fz, Cz, and Pz) were placed on the scalp, which were referenced to computer-linked earlobe electrodes [14]. The sampling frequency was set as 256 samples per second. Participants lay quietly in a sound-attenuated and less intrusive room at least 15 h during EEG recording. 

### 2.3. EEG Preprocessing 

Custom MATLAB scripts, along with the fieldtrip toolbox, were used for all signal processing. The data preprocessing method was modified from a previous study of patients with epilepsy [15]. Preprocessing of the EEG data consisted of line noise removal (2 Hz bandstop filters centered at 50 Hz with harmonics up to 100 Hz). This was followed by an automated artifact rejection algorithm: For each 30-s stage, each time point was converted into a z-score, based on the participant- and stage-specific mean and standard deviation of absolute amplitude, gradient (amplitude difference between two adjacent time points), and amplitude after application of the bandpass filter (0.5 Hz–70 Hz). A time point was then marked as artifactual if it exceeded either a z-score of 5 in any of these measures or the combination of an amplitude z-score of 3 and a gradient- or high-frequency z-score of 3. The 1 s preceding as well as the 1 s following the detected artifact samples were also marked as artifacts. 

### 2.4. KLDs Calculation

For human EEG, the sleep stages were first classified by visual evaluation of EEG signals from multiple leads by the experimenter. We then analyzed the custom codes based on the standards of the American Academy of Sleep Medicine (AASM) manual for adult sleep [16]. Briefly, the 30s epochs with low delta power or high alpha EEG power in the O2 leads were recognized as awake states. NREM epochs were identified as high delta power associated with lower-to-awake beta power. The NREM epochs with enhanced spindle activities in the C3 lead were classified as N2 stages, and the remaining NREM epochs with larger or smaller delta powers than the N2 stage were classified as the N3 and N1 stages, respectively. Awake and NREM stages were used in the following calculations (Figure 1B).

In the current study, the Kullback–Leibler divergence (*KLD*, relative entropy) was applied to measure the difference between one probability distribution and another reference probability distribution to compare day versus nighttime EEG. After filtering with the Butterworth filter, the energy of each 10 s epoch of each EEG band was calculated. The resulting epoch powers were classified into two groups, awake/NREM pair, and the respective probability distributions were calculated. The Kullback–Leibler divergence was calculated using the following formula [17,18]:$$KLD(P_{f,s2}||Q_{f,\;s1})=\underset{i}{\sum }Q_{f,\;s1}\left(i\right)ln\frac{Q_{f,\;s1}\left(i\right)}{P_{f,\;s2}\left(i\right)}$$
where *Q* represents the distribution of the absolute EEG activity at a particular band f under state *s*1, and *P* indicates the distribution of EEG activity at the same frequency band f in the other state *s*2.

### 2.5. Statistical Analyses

Data were expressed as the mean value ± standard deviation for continuous variables, and categorical variables were expressed as numbers. The two-sided independent samples t-test was applied to test the significant differences between the groups for normal distribution. For parameters that failed the normal distribution test, the Mann–Whitney U test was performed instead. The chi-square test was used for categorical comparisons of nominal values. The correlation between the HAMD-17 scores and the KLDs of high-frequency bands in specific brain areas was evaluated using Spearman’s correlation test. Statistical significance with *p* value lower than 0.05 was admitted for all statistical results. The data were analyzed with SPSS Version 21.0 (SPSS, Chicago, IL, USA).

## 3. Results

### 3.1. Baseline Characteristics of the Participants

Based on the established criteria, a total of 64 participants were included in the study. Out of the 64 eligible patients, 36 (56.2%) were females and 28 (43.8%) were males, with a mean age of 36.3 (SD: 14.5). The most common type of epilepsy in this population was focal epilepsy (68.7%), followed by generalized epilepsy (17.2%) and the remaining unknown epilepsy types (14.1%). This distribution was similar to that of a previous study [19]. Patients were diagnosed at a median age of 26.7 and had a disease duration of 7.5 years. Forty-eight patients (75%) were not taking anti-seizure medicines (ASMs) or took one drug, while the other sixteen patients (25%) took at least two drugs. The demographic and clinical characteristics of the patients are detailed in Table 1.

### 3.2. Comparison of Clinical Characteristics between Patients with and without Depression

Of the total 64 patients with epilepsy, 32 were in the depression group (mean age 37.0 ± 15.6, HAMD-17 score > 7) and the other 32 were in the non-depression group (mean age 35.3 ± 13.8, HAM-D score ≤ 7). To assess the possible role of demographic characteristics and epilepsy-related features in the presence of depression, a comparison of clinical characteristics between patients with and without depression was conducted. There were no significant differences in clinical characteristics, age, sex, latency, first diagnosis of epilepsy, type of seizure, and age at epilepsy diagnosis between the two groups. In addition, we did not find a significant difference in ASM usage between these two groups (details are shown in Table 2). To clarify the correlation between the depressive status and relevant variables, a correlation analysis was conducted between the HAMD-17 score and external variables. The results did not reveal a significant correlation between HAMD-17 score and age, gender, latency, epilepsy type, age at epilepsy diagnosis, ASMs, AVLT immediate recall, AVLT delayed recall, and AVLT recognition (*p* > 0.05, Appendix A). Moreover, a negative correlation was found between HAMD-17 and MMSE (r = −0.27, *p* = 0.04). This result is consistent with previous studies [20] and indicates that a reduction in cognitive function may be related to depressive disease in patients with epilepsy.

### 3.3. Global Analysis of KLDs between the Two Groups

The sleep stages of each participant were classified based on their long-term EEG results. Because it was difficult to visually distinguish the difference between these two groups (Appendix A), we calculated the circadian variation of EEG oscillation between awake and NREM stages in each brain region. We found that the KLDs of the high-frequency band (>30 Hz) in the depression group was significantly decreased in all brain regions, except for the parietal lobe and parietal-occipital lobe, and this difference was particularly pronounced in the frontal lobe (Figure 2). No significant difference in the low-frequency band was found between the two groups in nearly all global areas. In addition, for both the depression and non-depression groups, the KLD curves were symmetrical between the left and right hemispheres. This suggests that the EEG rhythms in the two hemispheres were similar between the corresponding regions, such as F3 and F4, F7 and F8, and Fp1 and Fp2.

### 3.4. Correlation Analysis between Regional-Specific KLDs and HAMD-17 Score

A detailed analysis was performed in the right frontal region (F4) because of the significant difference in the high-frequency bands in the preliminary analysis. We found that the KLDs for the gamma bands were significantly decreased in the depression groups (HAMD-17 score > 7) compared to the non-depression group (HAMD-17 score ≤ 7; KLD_D_ = 0.35 ± 0.05; KLD_ND_ = 0.57 ± 0.05; *p* = 0.009; Figure 3). We then modeled the relationship between the KLDs at different frequency bands and the HAM-D scores using a linear correlation analysis. The KLDs at the gamma band (30–40 Hz) were negatively correlated with participants’ HAMD-17 scores, with a correlation coefficient −0.31 (*p* = 0.01; Figure 4). These data indicate that regional-specific circadian differences reflect depressive status in patients with epilepsy.

## 4. Discussion

In this study, we found that patients with depressive symptoms had a significant sleep-wake variation at high-frequency oscillations in most of brain areas, especially in the frontal lobe. The KLDs at the gamma band (30–40 Hz) in the F4 area were negatively correlated with HAMD-17 scores in the further analysis. These findings may be of interest in helping us to better understand the EEG features in patients with epilepsy who have depressive symptoms.

Advancements in the understanding of EEGs have made them a powerful tool for the evaluation of many disorders, including delirium, psychiatric syndromes, and dementia [21]. In the present study, the utility of scalp EEG in assessing sleep-wake rhythms also exhibited a strong value. The KLD index was firstly established for awake and NREM EEG. Given the higher incidence of depression in patients with epilepsy, it seemed important to evaluate whether scalp EEG has potential value for assessing depression in patients with epilepsy. Interestingly, our study demonstrated that patients with depression had decreased high-frequency band oscillations in the frontal cortex relative to those without depression. These results echo prior studies reporting that frontal EEG asymmetry or higher frontal band power distinguishes major depressive disorder (MDD) from healthy controls [22,23,24]. It has been proven that frontal lesions and reduced frontal blood flow can lead to slow responses, loss of memory, and worse moods, which are manifestations of depression [25]. These findings are paralleled by metabolic alterations revealed by positron emission tomography studies that show decreased glucose metabolism in prefrontal areas [26]. Moreover, in a study on functional magnetic resonance imaging, altered blood-oxygen-level-dependent signals primarily in the prefrontal region were also found in MDD patients compared with healthy participants [27]. In addition to the frontal lobe, which is known as a locus for emotion and is involved in the pathogenesis of depression, there is another possible mechanism. Abnormalities of the frontal cortex are present, so function is very unlikely to be ‘‘localised’’; instead, to implicate an area of the frontal cortex is to implicate an abnormal network of emotional connections [28]. The processing and regulation of emotion is based on cortical-subcortical interactions, described as a neural network, rather than on one single brain region. This network mainly includes the frontal cortex, thalamus, and amygdala, which are involved in almost all functions related to emotion [29,30,31]. Not only was the thalamus an important structure for emotional regulation, it also involved in sleep-wake processes [32]. Meanwhile, the functional connections between the thalamus and frontal lobe are also sensitive to even mild sleep-wake variations [33]. Sleep-wake abnormalities can cause functional changes in the thalamic-emotional core region, which may lead to depression occurrence mediated by emotional neural networks [31,34]. In our study, the frontal lobe is the region where the circadian changes were most significantly decreased. Thus, the frontal lobe and its emotional network may serve as candidate targets for future stimulation intervention.

In addition, research has found that participants with MDD had a decreased frontal cortex gamma in emotion-related tasks [35]. Treatment with antidepressant-dose ketamine is rapid and long-lasting in helping patients with depression. Human studies have possibly implied that ketamine’s mechanism of action as an antidepressant may be related to an increase in gamma rhythm power [36]. Acute systemic administration of ketamine in rodents can enhance gamma power in a variety of brain regions [37]. Gamma rhythms do indeed correlate with increased neuronal action potential generation, and they are related to sensory systems and mood swings [24]. Optimal gamma rhythms may reflect a functional optimal balance between excitation and inhibition in key microcircuits [38]. These findings are similar to our results that patients with depression displayed circadian changes in EEG oscillations for gamma bands when compared with those without depression. This yields a novel possibility of using gamma power in the prefrontal cortex as a biomarker, because this brain region is heavily implicated in emotional regulation. Furthermore, this may also improve therapeutics via altering gamma rhythms themselves. However, discussing such therapy is beyond the scope of this work. Future research should more closely investigate the diagnosis and treatment value of the KLD of high-frequency band oscillations for patients with depression by considering the role of the possibly involved mechanisms. In addition, patients with epilepsy often have psychiatric comorbidities, of which sleep disorder and depression are the two most common [7,39]. Sleep disturbance and depression are often comorbid, and they possess underlying neurobiological correlates [40]. Sleep disturbance and depression are not only clinically related, but they are also genetically related with a correlation coefficient of 0.64 [41]. Previous studies found that multiple transcriptional-translational feedback loops controlled circadian rhythms throughout the SCN composed by molecular clock genes. The abnormal expression of clock genes is assumed to be an important factor in the occurrence of both insomnia and depression [9]. As is known, epilepsy tends to disrupt circadian rhythm. Apart from comorbid sleep disorder, abnormal circadian rhythms caused by epilepsy may lead to abnormal mood symptoms, which explains our finding that abnormal sleep-wake rhythms have a close relationship with depressive symptom in patients with epilepsy.

As far as we are aware, this is the first study that systematically quantified specific EEG features as an index of sleep-wake patterns from long-term scalp EEGs, and investigated the relationship between KLD and depressive severity in patients with epilepsy. Participants in this study were a group of patients with untreated depression, which ensured the robustness of the conclusion. 

However, some limitations should also be considered before generalizing the present findings. Firstly, the total number of participants in this dataset was relatively small. Thus, we cannot avoid the possibility of type II errors. Secondly, it is known that EEGs can be affected by the antiepileptic drugs (e.g., carbamazepine, oxcarbazepine, valproate, and lamotrigine) [42]. Whether taking ASMs or not, and the number of ASM types at the time of examination were also recorded. Moreover, the considerable strength of our conclusions was demonstrated when the regimen baseline of ASM administration between two groups was performed and no significant difference was found in this study. Thirdly, it is hard to exclude whether confounders such as seizure frequency or other psychiatric co-morbidities have an influence in the KLD assessment of sleep-wake patterns [43,44], which may also affect the relationship between KLD and depressive severity. Therefore, future work should further confirm this interaction.

## 5. Conclusions

In summary, this study demonstrated the high-utility value of scalp EEGs in quantifying sleep-wake variation. Furthermore, a negative correlation between the KLDs of gamma-band oscillations and the HAMD-17 scores was found. However, we should remain cautious about whether the KLD index can be a valid clinical tool for evaluating depressive severity in patients with epilepsy as further research is needed to address the limitations of this study.

## Figures and Tables

**Figure 1 brainsci-13-00823-f001:**
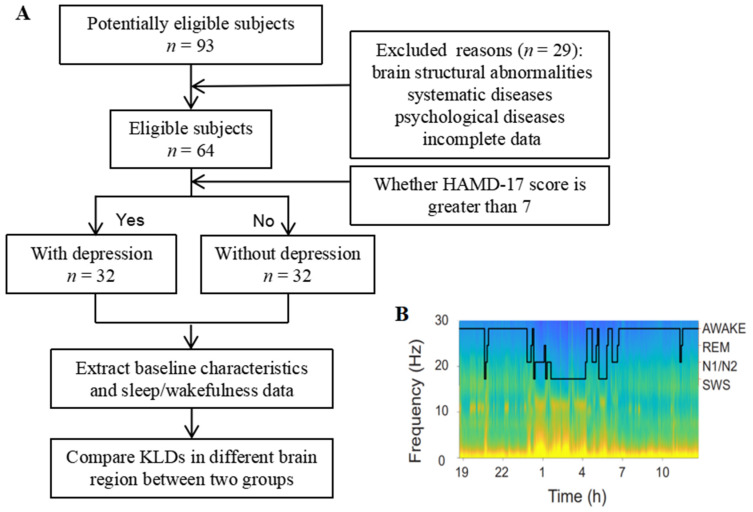
Experimental methods. (**A**) Flowchart of the study. (**B**) Spectrogram of EEG activity in a patient with epilepsy on this study. EEG powers in different frequency were coded with pseudo-color. Hypnogram (black line) was overlapped on the spectrogram. HAMD-17 = 17-item Hamilton Depression Scale, EEG = electroencephalogram, REM = rapid eye movement, SWS = slow wave sleep.

**Figure 2 brainsci-13-00823-f002:**
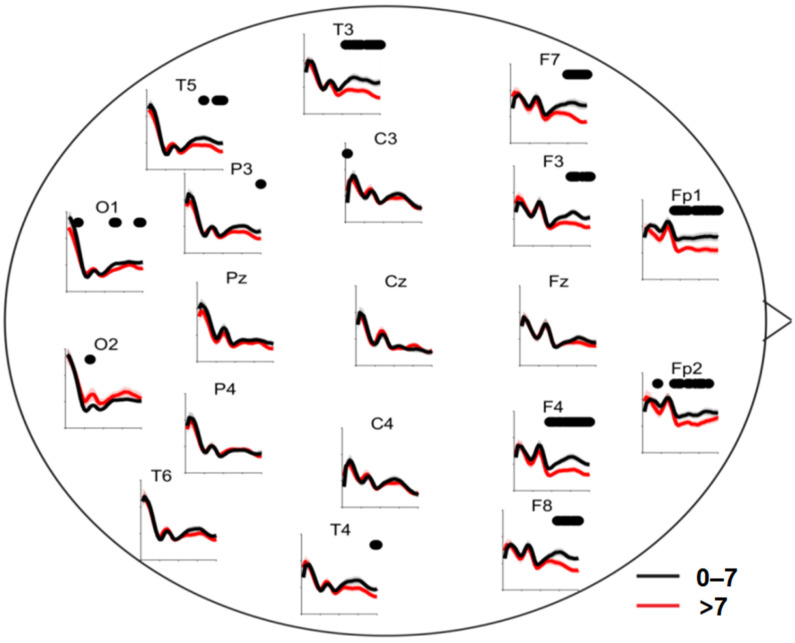
Circadian changes in EEG oscillations in global brain region between the two groups. Axes: KLDs (y-axis) curves vs. EEG frequency (x-axis). Red lines: KLDs of patients with HAMD-17 scores larger than 7. Black lines: KLDs of patients with HAMD-17 scores not larger than 7. Black dots on the top of axes: significant differences between KLDs of two groups at the frequency (*p* < 0.05, Student’s t-test), KLD = Kullback–Leibler divergence.

**Figure 3 brainsci-13-00823-f003:**
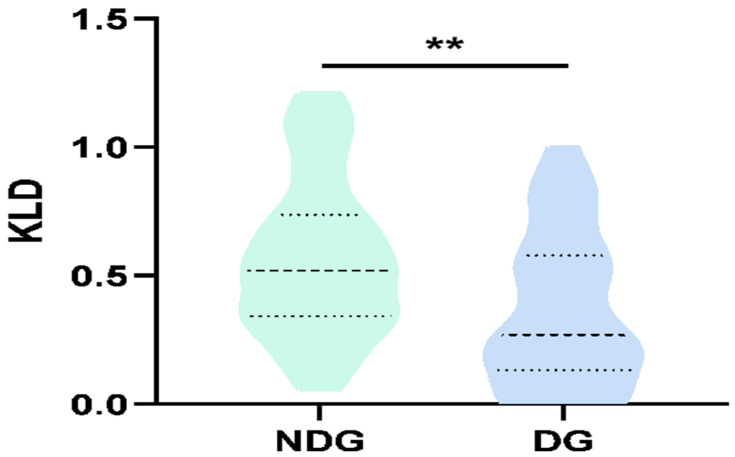
Relationship between KLDs of gamma-band (30–40 Hz) oscillations and HAMD-17 scores in F4 brain region. Significant differences of the KLDs at gamma band of patients with depression and without depression. (**, *p* < 0.01, DG = depression group, NDG = non-depression group).

**Figure 4 brainsci-13-00823-f004:**
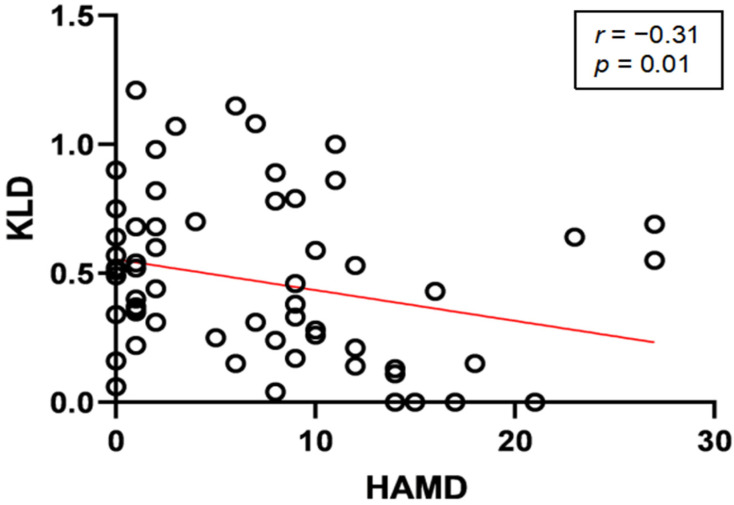
Relationship between KLDs of gamma (30–40 Hz) band oscillations and HAMD-17 scores in F4 brain region. Linear regression (red line) of KLDs at gamma band (30–40 Hz) vs. HAMD scores of patients. Black circles represent KLD vs. HAMD-17 score for each patient.

**Table 1 brainsci-13-00823-t001:** Demographic and clinical characteristics of the included patients.

Characteristics	*n* (%) or Mean ± SD/M (P25, P75)
Sex	
Female	36 (56.2)
Male	28 (43.8)
Age	36.3 ± 14.5
Epilepsy duration	7.5 (2.3, 15.8)
First diagnosis	
FD	14 (21.9)
NFD	50 (78.1)
Age at epilepsy diagnosis	26.7 ± 14.5
Epilepsy type	
Temporal lobe epilepsy	41 (64.1)
Frontal lobe epilepsy	2 (3.1)
Occipital lobe epilepsy	1 (1.5)
Generalized epilepsy	11 (17.2)
Unknown epilepsy	9 (14.1)
ASM numbers	
≤1	48 (75.0)
≥2	16 (25.0)
HAMD-17	7.5 (1.0, 11.8)
MMSE	29.0 (28.0, 30.0)
DS-forward	8.0 (8.0, 9.0)
DS-backward	5.2 ± 1.9
VFT-animals	16.2 ± 5.9
SDMT	48.4 ± 12.9
AVLT immediate recall	19.5 ± 5.4
AVLT delayed recall	8.0 ± 2.5
AVLT recognition	12.0 (11.0, 12.0)

Note: FD = first diagnosis, NFD = non-first diagnosis, ASM = antiseizure medicines, HAMD-17 = 17-item Hamilton Depression Rating Scale, MMSE = Mini-Mental State Examination, AVLT = Rey Auditory Verbal Learning Test, DS = digital span, SDMT = Symbol Digit Modalities Test, VFT = verbal fluency test, SD = standard deviation, M = median.

**Table 2 brainsci-13-00823-t002:** Comparison of baseline characteristics between patients with depression and without depression.

Variable	Without Depression	With Depression	*p*-Value
	(32)	(32)	
Sex (male/female)	17/15	11/21	0.13
Age (x ± s)	35.3 ± 13.8	37.0 ± 15.6	0.66
Epilepsy duration M (P_25_, P_75_)	6.5 (2.3, 18.3)	9.5 (2.3, 15.0)	0.76
First diagnosis (FD/NFD)	7/25	7/25	1.00
Age at epilepsy diagnosis (x ± s)	26.2 ± 14.0	27.6 ± 14.9	0.70
Epilepsy type (*n*)			0.15
Temporal lobe epilepsy (*n*)	25	16	
Frontal lobe epilepsy (*n*)	0	2	
Occipital lobe epilepsy (*n*)	0	1	
Generalized epilepsy (*n*)	4	7	
Unknown epilepsy (*n*)	3	6	
ASM numbers (*n*)			0.56
≤1	25	23	
≥2	7	9	
ASM types (*n*)			
CBZ	7	6	0.76
VA	8	9	0.78
OXC	6	2	0.26
LA	7	4	0.32
PB	1	1	1.00
TPM	1	5	0.20
LEV	4	2	0.67
PNT	1	1	1.00
CZP	0	5	0.06
HAMD-17 M (P25, P75)	1.0 (0.0, 2.0)	11.5 (9.0, 15.0)	0.00 ***
MMSE M (P25, P75)	28.0 (28.0, 29.0)	29.0 (28.0, 30.0)	0.02 *
DS-forward	8.0 (8.0, 9.0)	8.0 (7.0, 9.0)	0.98
DS-backward	5.6 ± 1.6	4.8 ± 2.2	0.11
VFT-animals	16.9 ± 6.9	15.5 ± 4.9	0.37
SDMT	52.4 ± 6.8	44.6 ± 16.0	0.05
AVLT immediate recall (x ± s)	18.4 ± 6.2	20.4 ± 4.4	0.17
AVLT delayed recall (x ± s)	7.6 ± 3.1	8.5 ± 2.0	0.20
AVLT recognition M (P25, P75)	12.0 (10.5, 12.0)	11.5 (11.0, 12.0)	0.93

Note: FD = first diagnosis, NFD = non-first diagnosis, ASM = antiseizure medicine, VA = valproate, LA = lamotrigine, LEV = levetiracetam, CBZ = carbamazepine, PNT = phenytoin, OXC = oxcarbazepine, CZP = clonazepam, TPM = topiramate, PB = phenobarbitone, HAMD-17 = 17-item Hamilton Depression Rating Scale, MMSE = Mini-Mental State Examination, AVLT = Rey Auditory Verbal Learning Test, DS = digital span, SDMT = Symbol Digit Modalities Test, and VFT = verbal fluency test, * *p* < 0.05, *** *p* < 0.001.

## Data Availability

The data underlying this article are available in the article and in its online Appendix A. The data will be shared on reasonable request to the corresponding author.

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
