# Peer review of "Kullback–Leibler Divergence of Sleep-Wake Patterns Related with Depressive Severity in Patients with Epilepsy"

_brainsci, 2023, doi:10.3390/brainsci13050823_

Round 1

Reviewer 1 Report

Comments:

This article is about use of Kullback-Libeler divergence of sleep-wake pattern in patients with depression and epilepsy. The research question is interesting. The manuscript is well written. Following points needs to be addressed:

1-     What was the study design? Authors mentioned that it was a retrospective study (Line 74). But no information regarding how the data were extracted, who were involved in the data extraction, how the EEG were assessed, who were the assessors, was blinding done etc has been provided.

2-     No mention of study duration in the manuscript.

3-     Were the clinical and neuropsychological data collected from database or clinical records?

4-     Was the neuropsychological assessment done by a single individual or not? How the investigators addressed the assessor variability?

5-     What measures were undertaken to address the possibility of bias in assessment of sleep -wake cycle?

6-     If I am correct, the EEG recordings were done for this study purpose. Not as a part of pre-surgical evaluation. Authors may clarify the same. If it is the later, then it should be mentioned that any ASM tapering was done or not.

7-     Confounders like ASM dosage, seizure frequency, other co-existent psychiatric co-morbidities has not been taken into account while assessing the KLD of sleep wake patterns. Authors should discuss them as potential limitations of their study.

8-     Full form of HAMD-17 should be mentioned while being mentioned first in the manuscript.

9-     Did the authors used any scale to evaluate the overall sleep quality and/or the sleep wake cycle in their study population?

10-  Use of sleep questionnaires, actigraphy or sleep diary might have provided useful sleep related data.

Minor corrections in english language are needed. Many typo errors are noted.

Author Response

Dear Reviewer:

Thank you for having our manuscript brainsci-2381610 “Kullback-Leibler Divergence of Sleep-Awake Patterns Related with Depressive Severity in Patients with Epilepsy” peer reviewed and we also thank you for giving us the opportunity to revise the manuscript for possible acceptance by the journal. We are now submitting a revised manuscript for your consideration of publication. In the revised manuscript, we have incorporated changes according to the criticisms by the reviewers. Any revisions to the manuscript have been marked up using the “Track Changes” function though MS Word. In addition, we have appended our Replies to Reviewers’ Comments. We believe that the revised manuscript has been greatly enhanced and would like to ask your kind consideration of acceptance of the manuscript.

If you or any of the reviewers has further questions, please do not hesitate to contact us.

Yours sincerely

Replies to Reviewers Comments

#1: What was the study design? Authors mentioned that it was a retrospective study (Line 74). But no information regarding how the data were extracted, who were involved in the data extraction, how the EEG were assessed, who were the assessors, was blinding done etc has been provided.

Reply: Thanks for the suggestion. This study was a retrospective study, and we have added information regarding how the data were extracted, who were involved in the data extraction, how the EEG were assessed, who were the assessors, was blinding done etc has been provided in first and second paragraph of methods section according to Reviewers' comment.

#2: No mention of study duration in the manuscript.

Reply: Thanks for the suggestion. We have added study duration in the first paragraph of methods section.

#3: Were the clinical and neuropsychological data collected from database or clinical records?

Reply: The clinical and neuropsychological data were collected from the database, and we have added related information in the manuscript.

#4: Was the neuropsychological assessment done by a single individual or not? How the investigators addressed the assessor variability?

Reply: Thanks for the comment. The neuropsychological assessment was done by a single trained investigator, which is less heterogeneous than several researchers.

#5. What measures were undertaken to address the possibility of bias in assessment of sleep-wake cycle?  

Reply: Regarding this comment, we analyzed the custom codes based on the standards of the American Academy of Sleep Medicine (AASM) manual for adult sleep, which was the cornerstone and principle of staging[1]. Briefly, the 30s epochs with low delta power or high alpha EEG power in the O2 leads were recognized as awake states. NREM epochs were identified as high delta power associated with lower-to-awake beta power. The NREM epochs with enhanced spindle activities in the C3 lead were classified as N2 stages, and the rest NREM epochs with larger or smaller delta powers than N2 stage were classified as the N3 and N1 stages, respectively. Thus, spectrum and power envelope in delta and spindle bands are produced to facilitate staging. This process is performed by two analysts. If there is a dispute result, the synchronous video was analyzed combined with the ictal event interference. Then the Kullback–Leibler divergence (KLD, relative entropy) was calculated on basis of the precise staging after the butterworth filter, so that the sleep-wake cycle can be evaluated more precise, thus reducing the possibility of bias.

#6.  If I am correct, the EEG recordings were done for this study purpose. Not as a part of pre-surgical evaluation. Authors may clarify the same. If it is the later, then it should be mentioned that any ASM tapering was done or not.  

Reply: Thanks for the comments. This study was a retrospective study and we extracted the EEG data from clinical database, thus, EEG recordings were not done only this study purpose, but a routine examination for variety of purposes such as diagnosis, efficacy assessment and discontinuation. Among the participants, there were patients with epilepsy diagnosed for the first time, as shown in Table 1 (FD accounted for 21.9%) and the patients diagnosed for the first time had not taken ASM. In previously diagnosed patients, the disease course was different (see row 3 in Table 1), so the amount, type, and dosage of ASM were also different. Fortunately, whether taking ASMs and the number and type of ASM at the time of examination have also been recorded. Moreover, the considerable strength of our conclusions was demonstrated when the regimen of ASM administration between two groups had no significant difference in this study.

#7. Confounders like ASM dosage, seizure frequency, other co-existent psychiatric co-morbidities has not been taken into account while assessing the KLD of sleep wake patterns. Authors should discuss them as potential limitations of their study. 

Reply: The problem mentioned exactly existed. The potential influence of ASM dosage, seizure frequency, other psychiatric co-morbidities on the KLD assessment of sleep-wake patterns have been discussed in limitation portion according to Reviewers' comment (the last paragraph of the Discussion section). This advise will be huge treasures for our future scientific activity.

#8. Full form of HAMD-17 should be mentioned while being mentioned first in the manuscript.

Reply: Thanks for pointing out our mistake. We have carefully added the full form of HAMD-17 while being mentioned first in the manuscript.

#9. Did the authors used any scale to evaluate the overall sleep quality and/or the sleep wake cycle in their study population?

Reply: The study population included from the database did not assessed overall sleep quality and/or sleep wake cycles through related scales. Although previous studies have reported that patients with epilepsy are prone to sleep disturbances and sleep-wake cycle disturbances[2,3], however, our study also needed to analyze the overall sleep quality and/or sleep-wake cycle of the population to better clarify the value of KLD. Therefore, our future work will further confirm this concern.

#10. Use of sleep questionnaires, actigraphy or sleep diary might have provided useful sleep related data.

Reply: Regarding this problem, there are some reasons why we chose EEG over sleep scales or actigraphs. Sleep diaries and sleep questionnaires are convenient, admittedly, these measures are subjective to some extent and may be deviated from the real situation. In clinical practice, the results were skewed by different evaluators, the recording way and assessment numbers[4]. Actigraphy has been gradually applied in clinical and animal studies in recent years. Actigraphy can indirectly evaluate sleep by recording the acceleration in the triaxial direction, and avoid invasive damage and allow a long following-up. However, this method cannot directly reflect sleep waves and distinguish sleep structures, such as NREM and REM phase[5]. PSG or EEG not only directly reflect sleep brainwaves, but also analyze each sleep wake stage and microstructure in detail. Currently, PSG or long-range EEG is the gold standard for the assessment of sleep-related problems[6]. Moreover, long range EEG has become a routine test for epileptic disease, such as diagnosis, efficacy evaluation and preoperative localization. These EEG data are readily available and highly objective[7].

Reference:

[1] Berry RB, Budhiraja R, Gottlieb DJ, et al. Rules for Scoring Respiratory Events in Sleep: Update of the 2007 AASM Manual for the Scoring of Sleep and Associated Events. J Clin Sleep Med 2012;8(5):597-619.

[2] Matos G, Andersen ML, do Valle AC, et al. The relationship between sleep and epilepsy: evidence from clinical trials and animal models. J Neurol Sci 2010;295:1–7.

[3] Mekky JF, Elbhrawy SM, Boraey MF, et al. Sleep architecture in patients with Juvenile Myoclonic Epilepsy. Sleep Med 2017;38:116–21.

[4] Odonnell S, Beaven CM, Driller CM, et al. From pillow to podium: a review on understanding sleep for elite athletes [J]. Nat Sci Sleep 2018;10:243-253.

[5] Meltzer LI, Hiruma LS, Avis K, et al. Comparison of a commercial accelerometer with polysomnography and actigraphy in children and adolescents [J]. Sleep 2015;38(8): 1323-1330.

[6] Stepnowsky C, Levendowski D, Popovic D, et al. Scoring accuracy of automated sleep staging from a bipolar electroocular recording compared to manual scoring by multiple raters [J]. Sleep Med 2013;14(11):1199-1207.

[7] Fisher RS, Cross JH, French JA, et al. Operational classification of seizure types by the International League Against Epilepsy: Position Paper of the ILAE Commission for Classification and Terminology. Epilepsia 2017;58(4):522-530.

Reviewer 2 Report

I appreciate your work, the study demonstrated the utility value of scalp EEGs. I appreciate that the limitations and strengths of the study have been incorporated. A section should be included where the tests used for the neuropsychological assessment of patients are explained.

It would be advisable to divide the Methods paragraph into two sub-paragraphs, one in which some characteristics of the sample are mentioned, for example number of participants, average age, male-female distribution, exclusion criteria. Another subparagraph should instead be dedicated to the Measures, in which a brief description of the neuropsychological tests considered in the study is provided and the corresponding bibliography shoulds be cited. Finally, the acronyms SD and M should be reported in full under Table 1. 

Author Response

Dear Reviewer:

Thank you for having our manuscript brainsci-2381610 “Kullback-Leibler Divergence of Sleep-Awake Patterns Related with Depressive Severity in Patients with Epilepsy” peer reviewed and we also thank you for giving us the opportunity to revise the manuscript for possible acceptance by the journal. We are now submitting a revised manuscript for your consideration of publication. In the revised manuscript, we have incorporated changes according to the criticisms by the reviewers. Any revisions to the manuscript have been marked up using the “Track Changes” function though MS Word. In addition, we have appended our Replies to Reviewers’ Comments. We believe that the revised manuscript has been greatly enhanced and would like to ask your kind consideration of acceptance of the manuscript.

If you or any of the reviewers has further questions, please do not hesitate to contact us.

Yours sincerely

#1: It would be advisable to divide the Methods paragraph into two sub-paragraphs, one in which some characteristics of the sample are mentioned, for example number of participants, average age, male-female distribution, exclusion criteria. Another subparagraph should instead be dedicated to the Measures, in which a brief description of the neuropsychological tests considered in the study is provided and the corresponding bibliography shoulds be cited. Finally, the acronyms SD and M should be reported in full under Table 1. 

Reply: Thanks for the comments. We have divided the Methods paragraph into two sub-paragraphs according to Reviewers' advise. In first sub-paragraph, some characteristics of the participants are mentioned while the detailed description of the neuropsychological tests is provided in the second sub-paragraph. Moreover, thanks for pointing out our mistakes. We have carefully reported the full form of acronyms SD and M. These suggestions will be huge treasures for our future scientific activity.

Reviewer 3 Report

Dear Authors,

I have read your paper with interest. I have following remarks:

1 . Please describe epilepsies in your group more clinically (types of seizures, epileptic syndromes presence)

2. Could you provide data on actual pharmacotherapy of patients and report whether there is a difference in drugs used in depressed and non-depressed population?

3. What do you mean by "unknown epilepsy"

4. How have you excluded possibility of pseudo-epileptic seizures? That may be of high importance in depressive patients.

I believe the manuscript require miso-moderate English editing.

Author Response

Dear Reviewer:

Thank you for having our manuscript brainsci-2381610 “Kullback-Leibler Divergence of Sleep-Awake Patterns Related with Depressive Severity in Patients with Epilepsy” peer reviewed and we also thank you for giving us the opportunity to revise the manuscript for possible acceptance by the journal. We are now submitting a revised manuscript for your consideration of publication. In the revised manuscript, we have incorporated changes according to the criticisms by the reviewers. Any revisions to the manuscript have been marked up using the “Track Changes” function though MS Word. In addition, we have appended our Replies to Reviewers’ Comments. We believe that the revised manuscript has been greatly enhanced and would like to ask your kind consideration of acceptance of the manuscript.

If you or any of the reviewers has further questions, please do not hesitate to contact us.

Yours sincerely

#1: Please describe epilepsies in your group more clinically (types of seizures, epileptic syndromes presence)

Reply: Thanks for the suggestion. We have detailedly described epilepsies more clinically in the first paragraph of Method section according to Reviewers' suggestion.

#2: Could you provide data on actual pharmacotherapy of patients and report whether there is a difference in drugs used in depressed and non-depressed population?

Reply: Thanks for the suggestion. Whether taking ASMs and the number and type of ASM at the time of examination have also been recorded. Furthermore, we analyzed the total ASM numbers and each ASM type between depressed and non-depressed population, and the results revealed considerable strength for final conclusion when the regimen of AED administration between two groups had no significant difference in this study. This results also have been showed in table 2. This advise will be huge treasure for our future scientific activity.

#3: What do you mean by "unknown epilepsy"

Reply: Thanks for the comment. Epilepsy type mainly includes generalized epilepsy, focal epilepsy, and unknown category according to the diagnostic criteria of the International League Against Epilpsy[1]. The term ‘unknown epilepsy’ is used to denote where it is understood that the patient has epilepsy but the clinician is unable to determine if the epilepsy type is focal or generalized because there is insufficient information available. This may be for a variety of reasons. There may be no access to EEG or the EEG studies may have been uninformative e.g. normal. If the seizure type(s) are unknown, then the epilepsy type may be unknown for similar reasons although the two may not always be concordant. For example, the patient may have had several symmetrical tonic-clonic seizures without focal features and normal EEG recordings. Thus, the onset of the seizures is unknown and the person has an unknown epilepsy type. We also have fully described the definition and example of unknown epilepsy in the first paragraph of method section.

#4: How have you excluded possibility of pseudo-epileptic seizures? That may be of high importance in depressive patients.

Reply: We have excluded possibility of pseudo-epileptic seizures. The protocols included a clinical examination and history-taking by trained neurologists. Additional complementary measures including video-EEG monitoring, MRI were performed. According to the diagnostic criteria of the International League Against Epilepsy, epilepsy was diagnosed based on clinical symptoms and electrical discharge in EEG combined by auxiliary imaging. In total, there are strict diagnostic criteria for epilepsy disease. Pseudo-epileptic seizures is caused by a variety of psychological and psychiatric reasons. The main feature is the presence of pseudoconvulsions or other abnormal behavior, but no epileptiform discharge on the EEG. Combined with the psychological scale and prolactin level assessment, it is not difficult to distinguish two diagnosis. Moreover, the epilepsy diagnosis of each included patient is definite, and these subjects are intended to take or has taken ASMs. 

Reference:

[1] Scheffer IE, Berkovic S, Capovilla G, et al. ILAE classification of the epilepsies: Position paper of the ILAE Commission for Classification and Terminology. Epilepsia 2017;58(4):512-521. 

Round 2

Reviewer 1 Report

Authors have addressed my queries adequately and necessary changes in the manuscript has been made. 

Minor editing of English language may be required.